# Reactive Oxygen Species-Dependent Activation of EGFR/Akt/p38 Mitogen-Activated Protein Kinase and JNK1/2/FoxO1 and AP-1 Pathways in Human Pulmonary Alveolar Epithelial Cells Leads to Up-Regulation of COX-2/PGE_2_ Induced by Silica Nanoparticles

**DOI:** 10.3390/biomedicines11102628

**Published:** 2023-09-25

**Authors:** Yan-Jyun Lin, Chien-Chung Yang, I-Ta Lee, Wen-Bin Wu, Chih-Chung Lin, Li-Der Hsiao, Chuen-Mao Yang

**Affiliations:** 1Institute of Translational Medicine and New Drug Development, College of Medicine, China Medical University, Taichung 40402, Taiwan; u108306002@cmu.edu.tw; 2Department of Traditional Chinese Medicine, Chang Gung Memorial Hospital at Tao-Yuan, Kwei-San, Tao-Yuan 33302, Taiwan; r55161@cgmh.org.tw; 3School of Traditional Chinese Medicine, College of Medicine, Chang Gung University, Kwei-San, Tao-Yuan 33302, Taiwan; 4School of Dentistry, College of Oral Medicine, Taipei Medical University, Taipei 11031, Taiwan; itlee0128@tmu.edu.tw; 5School of Medicine, Fu Jen Catholic University, New Taipei City 242062, Taiwan; wenbin@mail.fju.edu.tw; 6Graduate Institute of Biomedical and Pharmaceutical Science, Fu Jen Catholic University, New Taipei City 242062, Taiwan; lidesiao@livemail.tw; 7Department of Anesthetics, Chang Gung Memorial Hospital at Linkuo Branch, Kwei-San, Tao-Yuan 33305, Taiwan; chihchung@adm.cgmh.org.tw; 8Department of Pharmacology, College of Medicine, China Medical University, Taichung 40402, Taiwan

**Keywords:** silica nanoparticles, COX-2, PGE_2_, alveolar epithelial cells, ROS, FoxO1

## Abstract

The risk of lung exposure to silica nanoparticles (SiNPs) and related lung inflammatory injury is increasing with the wide application of SiNPs in a variety of industries. A growing body of research has revealed that cyclooxygenase (COX)-2/prostaglandin E_2_ (PGE_2_) up-regulated by SiNP toxicity has a role during pulmonary inflammation. The detailed mechanisms underlying SiNP-induced COX-2 expression and PGE_2_ synthesis remain unknown. The present study aims to dissect the molecular components involved in COX-2/PGE_2_ up-regulated by SiNPs in human pulmonary alveolar epithelial cells (HPAEpiCs) which are one of the major targets while SiNPs are inhaled. In the present study, we demonstrated that SiNPs induced COX-2 expression and PGE_2_ release, which were inhibited by pretreatment with a reactive oxygen species (ROS) scavenger (edaravone) or the inhibitors of proline-rich tyrosine kinase 2 (Pyk2, PF-431396), epidermal growth factor receptor (EGFR, AG1478), phosphatidylinositol 3-kinase (PI3K, LY294002), protein kinase B (Akt, Akt inhibitor VIII), p38 mitogen-activated protein kinase (MAPK) (p38 MAPK inhibitor VIII), c-Jun N-terminal kinases (JNK)1/2 (SP600125), Forkhead Box O1 (FoxO1, AS1842856), and activator protein 1 (AP-1, Tanshinone IIA). In addition, we also found that SiNPs induced ROS-dependent Pyk2, EGFR, Akt, p38 MAPK, and JNK1/2 activation in these cells. These signaling pathways induced by SiNPs could further cause c-Jun and FoxO1 activation and translocation from the cytosol to the nucleus. AP-1 and FoxO1 activation could increase COX-2 and PGE_2_ levels induced by SiNPs. Finally, the COX-2/PGE_2_ axis might promote the inflammatory responses in HPAEpiCs. In conclusion, we suggested that SiNPs induced COX-2 expression accompanied by PGE_2_ synthesis mediated via ROS/Pyk2/EGFR/PI3K/Akt/p38 MAPK- and JNK1/2-dependent FoxO1 and AP-1 activation in HPAEpiCs.

## 1. Introduction

Nanotechnology has numerous applications in biology and medicine, although the beneficial and harmful effects of nanoparticles on human health and the environment have sparked controversy [1,2]. Silica, or silicon dioxide (SiO_2_), consists of silicon and oxygen, the two most abundant elements in the Earth’s crust. Silica primarily exists in crystalline or amorphous forms. Among these, amorphous silica encompasses both natural and man-made sources [2,3]. Amorphous silica nanoparticles (SiNPs) are defined as nanosized SiO_2_ materials measuring less than 100 nm. Due to their highly adaptable biocompatibility and stability, SiNPs find wide-ranging applications in various fields, including cosmetic additives, printer toners, dietary supplements, and biomedical applications such as drug delivery, gene carriers, molecular imaging, and cancer therapy [2,3]. However, previous research reports have suggested that SiNPs may pose more significant risks to human health, particularly the respiratory system, compared to crystalline silica [4]. Nevertheless, the precise impact of SiNPs on respiratory health and the detailed underlying pathogenic mechanisms remain to be elucidated. Notably, nanoparticulate matter can readily affect health through inhalation [5,6]. Consequently, pulmonary alveolar epithelial cells represent a major target of SiNPs. In recent years, mounting evidence has demonstrated that SiNPs exert toxic effects on the airway, manifesting as cytotoxicity and inflammatory responses [5,6,7]. Previous studies have indicated that exposure to SiNPs induces the expression of cyclooxygenase (COX)-2 in human lung epithelial cells [7]. However, the precise mechanisms underlying SiNP-induced inflammatory responses in human pulmonary alveolar epithelial cells (HPAEpiCs) remain unclear. The present study aims to unravel the molecular mechanisms responsible for SiNP-induced COX-2 expression in HPAEpiCs.

COX is the key enzyme responsible for the transformation of arachidonic acid into prostaglandins (PGs), which exist in two isoforms: COX-1 and COX-2 [8]. COX-1 is constitutively expressed in most cells throughout the body and plays a crucial role in maintaining homeostatic functions. In contrast, the expression of COX-2 is highly inducible in various inflammatory situations, leading to an up-regulated production of PGs [9]. Inflamed tissues exhibit a significant increase in both the level and biosynthesis of PGs [10]. The up-regulation of COX-2 and PGs expression has been observed in inflammatory pulmonary diseases, making it a crucial pathogenic factor in conditions such as asthma, chronic obstructive pulmonary disease, and lung tumors [11,12,13]. Our previous studies have also shown that both COX-2 levels and PGE_2_ synthesis are up-regulated in the inflammatory model of the airways induced by pro-inflammatory mediators [14,15].

Reactive oxygen species (ROS) are pivotal signaling molecules that play a crucial role in the progression of inflammatory disorders [6]. Furthermore, SiNPs have been confirmed to elevate intracellular ROS production, leading to oxidative stress in various cell types, which subsequently promotes inflammation, DNA damage, and cell death [6,16,17,18,19]. Many studies have indicated that ROS play critical roles in regulating COX-2 expression through a variety of signaling pathways [20,21]. Protein tyrosine kinases (PTKs) serve as intermediaries in transducing extracellular signals to the cytoplasm and further mediate downstream effector pathways. Overexpression or activation of PTKs has been linked to the development of numerous diseases, including cancer, inflammation, cardiovascular conditions, and neurodegenerative disorders [22,23]. Previous studies have highlighted the involvement of epidermal growth factor receptor (EGFR) and the non-receptor tyrosine kinase, proline-rich tyrosine kinase 2 (Pyk2), in airway inflammatory responses [22], as well as the up-regulation of COX-2 [23]. The phosphatidylinositol 3-kinase (PI3K)/protein kinase B (Akt) pathway [22] and the mitogen-activated protein kinases (MAPKs) pathway (including extracellular signal-related kinases, ERKs, c-Jun N-terminal kinases, JNKs, and p38 mitogen-activated protein kinase, p38 MAPK) [24,25] are considered as the most common EGFR-mediated downstream signaling components in regulating lung inflammatory responses and further enhancing COX-2 and PGE_2_ production [15,26,27]. Various transcription factors, such as activator protein 1 (AP-1) [11,28] and Forkhead Box class O1 (FoxO1) [27], are known to regulate inflammatory responses and COX-2 expression. Furthermore, these transcription factors can be activated by EGFR, PI3K/Akt, or MAPKs [11,27,28]. Therefore, we hypothesize that these signaling components and transcription factors are involved in SiNP-induced COX-2 expression and PGE_2_ synthesis in HPAEpiCs. The present results suggest that SiNP-induced COX-2 and PGE_2_ expression are, at least in part, mediated through the ROS/Pyk2/EGFR/PI3K/Akt/p38 MAPK and JNK1/2 cascade-dependent activation of FoxO1 and AP-1 in HPAEpiCs.

## 2. Materials and Methods

### 2.1. Materials

SiNPs (nanopowder, particle size between 10 and 20 nm, # 637238) were purchased from Sigma (St. Louis, MO, USA). Dulbecco’s modified Eagle’s medium (DMEM)/F-12 medium, fetal bovine serum (FBS), TRIZOL reagent, 2′,7′-dichlorofluorescin diacetate (H_2_DCF-DA), and M-MLV Reverse Transcriptase kit were purchased from Invitrogen (Carlsbad, CA, USA). BioTrace™ NT nitrocellulose transfer membrane was from Pall Corporation (Port Washington, NY, USA). Enhanced chemiluminescence reagent was from EMD Millipore Corporation (Burlington, MA, USA). GenMute^TM^ siRNA Transfection Reagent was obtained from SignaGen Lab (Gaithersburg, MD, USA). Actinomycin D (Act. D), cycloheximide (CHI), edaravone, AG1478, PF-431396, LY294002, Akt inhibitor (Akti) VIII, p38 MAPK inhibitor (p38i) VIII, SP600125, AS1842856, tanshinone IIA were purchased from Biomol (Plymouth Meeting, PA, USA). Anti-COX-2 (#12282S), anti-GAPDH (#2118L), anti-phospho-Pyk2 (#3291S), anti-phospho-EGFR (Tyr^1173^, #4407L), anti-P110 (#4249S), anti-phospho-Akt (Ser^473^, #9271L), anti-Akt (#4691L), anti-phospho-p38 MAPK (Thr^180^/Tyr^182^, #9211L), anti-p38 MAPK (#8690S), anti-phospho-SAPK/JNK (Thr^183^/Tyr^185^, #4668S), anti-phospho-FoxO1 (#9461S), anti-FoxO1 (#2880S), anti-phospho-c-Jun (#2361S), and anti-c-Jun (#9165L) antibodies were from Cell Signaling Technology (Danvers, MA). Anti-Pyk2 (#ab32448) antibody was from Abcam (Cambridge, UK). Anti-EGFR (sc-373746) and anti-JNK1/2 (sc-137020) antibodies were from Santa Cruz (Santa Cruz, CA, USA). Enzymes and other chemicals were from Sigma (St. Louis, MO, USA).

### 2.2. Cell Culture

HPAEpiCs were purchased from ScienCell Research Laboratories (San Diego, CA, USA) and cultured as previously described [29]. The cell suspension was diluted with DMEM/F-12 containing 10% FBS and plated onto 12-well culture plates (3 × 10^4^ cells/well) and 6-well culture plates (8 × 10^4^ cells/well) for the measurement of protein expression and mRNA accumulation. HPAEpiCs passages from 5 to 7 were used throughout this study.

### 2.3. Western Blot Analysis

Growth-arrested HPAEpiCs were incubated without or with different concentrations of SiNPs at 37 °C for the indicated time intervals. When pharmacological inhibitors were used, they were added for 1 h prior to the application of SiNPs. As previously described [29], the cells after incubation were rapidly washed, harvested, denatured by heating for 15 min at 95 °C, and centrifuged at 45,000× *g* at 4 °C to prepare the whole cell extract. The samples were subjected to SDS-PAGE using a 10% running gel and transferred to nitrocellulose membrane. The membrane was incubated successively overnight at 4 °C with one of the primary antibodies, and then incubated with 1:2000 dilution of an anti-rabbit or anti-mouse antibody for 1 h at room temperature. Following incubation, the membranes were washed extensively with TTBS. The immunoreactive bands were visualized by an enhanced chemiluminescence reagent. The images of the immunoblots were captured by a UVP BioSpectrum 500 imaging system (Upland, CA, USA), and densitometry analysis was executed by UN-SCAN-IT gel 7.1 software (Orem, UT, USA).

### 2.4. Real-Time PCR Analysis

Total RNA was isolated from HPAEpiCs in 6-well culture plates (8 × 10^4^ cells/well) treated with SiNPs for the indicated time intervals and extracted with 500 μL TRIzol. RNA concentration was spectrophotometrically determined at 260 nm/280 nm. As previously described [29], the cDNA obtained from 5 μg total RNA was used as a template for PCR amplification. The primers and probe mixtures were used for COX-2 and GAPDH. PCR was performed using a StepOnePlus™ Real-Time PCR System (Applied Biosystems™/Thermo Fisher Scientific, Foster City, CA, USA). The relative amount of the target gene was calculated using 2^(Ct test gene-Ct GAPDH)^ (Ct = threshold cycle). Oligonucleotide primers for human COX-2 and GAPDH were used as the follows:

COX-2 (NM_000963.4)

5′-CAAACTGAAATTTGACCCAGAACTAC-3′ (Sense)

5′-ACTGTTGATAGTTGTATTTCTGGTCATGA-3′ (Anti-sense)

5′-AACACCCTCTATCACTGGCATCCCCTTC-3′ (Probe)

GAPDH (NM_001357943.2)

5′-GCCAGCCGAGCCACAT-3′ (Sense)

5′-CTTTACCAGAGTTAAAAGCAGCCC-3′ (Anti-sense)

5′-CCAAATCCGTTGACTCCGACCTTCA-3′ (Probe)

### 2.5. Measurement of PGE_2_ Release

Growth-arrested HPAEpiCs were incubated without or with different concentrations of SiNPs at 37 °C for the indicated time intervals while pharmacological inhibitors were applied for 1 h prior to the treatment of SiNPs, the supernatants were collected to analyze PGE_2_ levels by using a PGE_2_ enzyme-linked immunosorbent assay (ELISA) kit (Enzo Life Sciences, Farmingdale, NY, USA) according to the product manual instructions.

### 2.6. Measurement of Intracellular ROS

Growth-arrested HPAEpiCs were treated with SiNPs for the indicated time intervals, or pretreated with pharmacological inhibitors for 2 h prior to the treatment of SiNPs, and then the culture medium was changed to warm PBS containing 5 μM H_2_DCF-DA for 20 min at 37 °C, as previously described [29]. The fluorescence intensity was measured by a fluorescence microplate reader (Synergy H1 Hybrid Reader, BioTek, VT, USA) with excitation/emission at 485/530 nm.

### 2.7. Transient Transfection with siRNAs

HPAEpiCs were cultured in 6-well culture plates at 80% confluence. SMARTpool RNA duplexes corresponding to c-Jun (HSS180003, HSS105641, HSS105642; NM_002228.4) and p38α (HSS102352, HSS102353, HSS175313; NM_001315.3) siRNAs were purchased from Invitrogen Life Technologies (Carlsbad, CA, USA), and Akt1 (SASI_Hs01_00105954; NM_005613), P110 (SASI_Hs01_00219339; NM_006218), JNK2 (SASI_Hs01_00143827; NM_002752), FoxO1 (SASI_Hs01_0076732; NM_002015), Pyk2 (SASI_Hs01_00032249; NM_004103), and scrambled siRNAs were obtained from Sigma-Aldrich (St. Louis, MO, USA). EGFR siRNA (SASI_Hs01_00215449; NM_005228) was purchased from Dharmacon, Inc. (Lafayette, CO, USA). Transient transfection of siRNAs (final concentration 100 nM) was formulated with GenMute™ siRNA transfection reagent according to the manufacturer’s instruction (SignaGen laboratories, Frederick, MD, USA), and then were directly added to the cells containing 900 μL of DMEM/F-12 medium at 37 °C for 15 h, as previously described [29]. The cells were washed with PBS and maintained in DMEM/F-12 containing 10% FBS for 10 h. Then cells were washed with PBS and incubated in serum-free DMEM/F-12 medium overnight before treatment with SiNPs for the indicated time intervals.

### 2.8. Cell Viability

For measurement of cell viability, HPAEpiCs were cultured in 12-well culture plates and made quiescent at confluence by incubation in serum-free DMEM/F-12 overnight. After treatment with SiNPs or pharmacological inhibitors, the viability of HPAEpiCs was determined by cell counting kit-8 [WST-8, 2-(2-methoxy-4-nitrophenyl)-3-(4-nitrophenyl)-5-(2,4-disulfophenyl)-2H-tetrazolium, monosodium salt] assay. The absorbance of samples was measured by a microplate reader (Bio-Rad, Hercules, CA, USA) with a wavelength of 450 nm.

### 2.9. Statistical Analysis of Data

All the data were expressed as the mean ± S.E.M. for at least three individual experiments (*n* = number of independent cell culture preparations). We applied GraphPad Prizm Program 6.0 software (GraphPad, San Diego, CA, USA) to statistically analyze, as previously described [29], by using one-way analysis of variance (ANOVA) followed by Tukey’s post hoc test. *p* values of 0.05 were considered to be statistically significant. Error bars were omitted when they fell within the dimensions of the symbols.

## 3. Results

### 3.1. SiNPs Induce COX-2 Expression and PGE_2_ Production in HPAEpiCs

COX-2/PGE_2_ is highly expressed in pulmonary inflammatory diseases [11,12,13]. To determine whether the harmful effects of SiNPs are through COX-2/PGE_2_, we evaluated COX-2/PGE_2_ expression under SiNP exposure in HPAEpiCs. The data depicted in Figure 1A indicating that SiNPs induced COX-2 protein expression in a time- and concentration-dependent manner. While the cells were exposed to 25 and 50 μg/mL SiNPs, a significant increase was observed within 2 h, and a maximal response was reached within 12 h. In addition, the data revealed that SiNP-induced COX-2 mRNA expression determined by real-time PCR was gradually up-regulated in a time-dependent manner, which was significantly increased within 6 h and reached a maximal response within 10 h (Figure 1B). To further ensure the COX-2 enzymatic activity under SiNP challenge in HPAEpiCs, PGE_2_ synthesis was determined. As shown in Figure 1C, SiNP-induced PGE_2_ synthesis was gradually presented with a maximal response within 12 h. In addition, exposure of HPAEpiCs to various concentrations of SiNPs showed that concentrations greater than 50 μg/mL significantly caused cell death (Figure 1D). Thus, the concentration of SiNPs at 25 μg/mL was used for the following experiments.

### 3.2. SiNPs Induce COX-2 Expression Mediated through Transcription and Translation in HPAEpiCs

To examine whether the mechanisms of SiNP-induced COX-2 expression and PGE_2_ secretion were via the transcriptional and translational levels, HPAEpiCs were stimulated with 25 μg/mL SiNPs with pretreatment of actinomycin D (Act. D, a transcriptional inhibitor) or cycloheximide (CHI, a translational inhibitor), respectively. Our findings in Figure 2A demonstrated that SiNP-induced COX-2 protein expression was dose dependently attenuated under either Act. D or CHI pretreatment. In the transcription regulation, we certified that Act. D significantly reduced SiNP-induced COX-2 mRNA expression in HPAEpiCs, but not CHI (Figure 2B). Comparable to the results of COX-2 protein expression, the PGE_2_ secretion was significantly repressed by Act. D and CHI pretreatment (Figure 2C). Taken together, these results suggested that SiNP-induced COX-2 expression and PGE_2_ synthesis are mediated through ongoing transcription and translation in HPAEpiCs.

### 3.3. ROS Generation Participates in SiNP-Induced COX-2 Expression and PGE_2_ Synthesis in HPAEpiCs

Overproduction of ROS is associated with inflammatory responses and has been thought to be involved in cell damage and cell death induced by nanoparticles [30,31]. We applied a ROS scavenger (edaravone) to pretreat HPAEpiCs prior to SiNP exposure. As shown in Figure 3A, SiNP-induced COX-2 protein expression was dose dependently attenuated by the pretreatment with edaravone. In addition, the increase of COX-2 mRNA levels was also attenuated by edaravone (Figure 3B). Furthermore, SiNPs time-dependently stimulated ROS generation with a maximal response within 30 min, which was reduced by pretreatment with edaravone (Figure 3C). Our data further verified that PGE_2_ secretion induced by SiNPs was also attenuated by the pretreatment with edaravone (Figure 3D). These results suggested that SiNPs mediate COX-2 up-regulation and PGE_2_ secretion through ROS generation in HPAEpiCs.

### 3.4. SiNPs Induce COX-2 Expression and PGE_2_ Synthesis through the Activation of Pyk2 in HPAEpiCs

The effect of Pyk2 is especially important in inflammatory diseases, which has been shown to play a key role in the development of pulmonary inflammation [32]. Therefore, we examined whether Pyk2 was involved in SiNP-induced responses. The cells were pretreated with PF-431396, a Pyk2 inhibitor, then challenged with SiNPs. As shown in Figure 4A, pretreating HPAEpiCs with PF-431396 significantly inhibited the SiNP-induced COX-2 expression in a concentration-dependent manner. Moreover, we revealed that SiNP-induced COX-2 mRNA expression was inhibited by PF-431396 pretreatment (Figure 4B). To further ensure the role of Pyk2 in COX-2 expression by SiNPs, cells were transfected with Pyk2 siRNA. While the Pyk2 protein was down-regulated, the level of COX-2 expression was simultaneously reduced in HPAEpiCs under SiNP stimulation (Figure 4C). We found that Pyk2 phosphorylation was involved in SiNP-induced response, which was attenuated by Pyk2 siRNA transfection or pretreatment of edaravone and NAC, respectively (Figure 4D). These findings indicated that Pyk2 was activated by an upstream component ROS in HPAEpiCs. Finally, the release of PGE_2_ induced by SiNPs was diminished by PF-431396 pretreatment (Figure 4E). These data suggested that SiNP-induced COX-2 expression and PGE_2_ secretion are mediated through activation of ROS-dependent Pyk2 cascade in HPAEpiCs.

### 3.5. EGFR Is Necessary for SiNP-Induced COX-2 Expression and PGE_2_ Synthesis

EGFR signaling has been reported to be involved in various cellular physiological and pathologic processes, especially in lung injury and inflammation [22,33,34]. Recent studies have shown that the EGFR signaling component may participate in the induction of COX-2 levels in lung cancer cell lines [35]. Thus, we used a selective EGFR inhibitor, AG1478, to investigate whether EGFR regulated SiNP-induced COX-2 expression. Our data found that AG1478 pretreatment reduced both SiNP-induced COX-2 protein and mRNA expression (Figure 5A,B). Then, we ensured these data by transfecting cells with EFGR siRNA. Based on data in Figure 5C, EGFR protein expression down-regulated by transfection with EGFR siRNA caused an obvious reduction of SiNP-induced COX-2 protein expression. In addition, EGFR phosphorylation was involved in the SiNP-induced response, which was attenuated by transfection with either EGFR or Pyk2 siRNA (Figure 5D). We also found that transfection with EGFR siRNA had no significant effects on Pyk2 phosphorylation. Moreover, inhibition of EGFR by AG1478 could reduce the SiNP-stimulated PGE_2_ secretion (Figure 5E). These results suggested that SiNP-induced COX-2 expression is mediated through Pyk2-dependent EGFR activation in HPAEpiCs.

### 3.6. PI3K/Akt Pathway Participates in SiNP-Induced COX-2 Expression and PGE_2_ Synthesis in HPAEpiCs

Growing evidence has unveiled that PI3K/Akt signaling pathway plays a crucial role in the regulation of lung inflammatory responses [22]. Thus, we tested the role of the PI3K/Akt signaling pathway in the SiNP-induced COX-2 expression in HPAEpiCs. For this purpose, we adopted the inhibitors of PI3K (LY294002) and Akt (Akt inhibitor Ⅷ). Data in Figure 6A showed that pretreatment with either LY294002 or Akt inhibitor Ⅷ significantly diminished the SiNP-induced COX-2 expression. In addition, LY294002 and Akt inhibitor Ⅷ also suppressed the SiNP-induced COX-2 mRNA expression (Figure 6B). To ascertain the roles of PI3K/Akt in the induction of COX-2 by SiNP stimulation, cells were transfected with siRNA of either p110 or Akt, respectively. While p110 and Akt proteins were down-regulated by their own siRNAs, the levels of SiNP-induced COX-2 protein were also inhibited in HPAEpiCs (Figure 6C). We further determined whether SiNP-induced response was mediated through Akt phosphorylation. As shown in Figure 6D, Akt phosphorylation induced by SiNPs was inhibited by transfection with siRNA of p110, Akt, or EGFR. We noted that transfection with Akt siRNA had no significant effects on SiNP-stimulated EGFR phosphorylation. Furthermore, SiNP-induced PGE_2_ production was inhibited by Akt inhibitor Ⅷ (Figure 6E). These results suggested that SiNP-induced COX-2 expression and PGE_2_ secretion are mediated through EGFR-dependent PI3K/Akt activation in HPAEpiCs.

### 3.7. SiNPs Induce COX-2 Expression through p38 MAPK Activation

The MAPK cascades regulate multiple intracellular signal functions in response to various extracellular stimuli [25]. Several pieces of evidence have unveiled that COX-2 induction is mediated through MAPK activation such as p38 MAPK and JNK1/2 [30]. We expected MAPKs to participate in the SiNP-induced COX-2 expression and PGE_2_ production in HPAEpiCs. We applied the pharmacologic inhibitor of p38 MAPK (p38 MAPK inhibitor VIII) to verify whether p38 MAPK has a role in the SiNP-induced COX-2 expression. We found that SiNP-stimulated COX-2 protein and mRNA expression were significantly repressed by p38 MAPK inhibitor VIII (Figure 7A,B). Then, p38α siRNA was used to ensure the role of p38 MAPK in the SiNPs-mediated COX-2 expression. As shown in Figure 7C, transfection with p38 siRNA reduced total p38 MAPK protein and SiNP-induced COX-2 expression. In addition, we evaluated whether p38 MAPK phosphorylation was involved in the SiNP-induced response in HPAEpiCs. Data in Figure 7D revealed that SiNPs induced phosphorylation of p38 MAPK in a time-dependent manner, which was attenuated by transfection with Akt or p38α siRNA. However, transfection with p38α siRNA had no significant effects on the SiNP-stimulated Akt phosphorylation. Moreover, we also observed that SiNP-induced PGE_2_ production was depressed by p38 MAPK inhibitor VIII (Figure 7E). Taken together, these results suggested that SiNP-induced COX-2 expression and PGE_2_ production are mediated through Akt-dependent p38 MAPK activation in HPAEpiCs.

### 3.8. SiNPs Induce COX-2 Expression and PGE_2_ Secretion via JNK1/2

As mentioned in the above evidence, JNK1/2 is involved in COX-2 induction. In addition, they also regulate SiNP-stimulated responses in various types of cells [30]. Next, we adopted the pharmacologic inhibitors of JNK1/2 (SP600125) to investigate whether JNK1/2 took part in the SiNP-induced COX-2 expression in HPAEpiCs. Data in Figure 8A,B demonstrated that SP600125 pretreatment inhibited the SiNP-induced COX-2 protein expression in a dose-dependent manner and COX-2 mRNA expression. Then, we used JNK2 siRNA transfection to verify the involvement of JNK1/2 in the SiNP-stimulated COX-2 expression in HPAEpiCs. As shown in Figure 8C, transfection with JNK2 siRNA knocked down total JNK2 protein expression and diminished SiNP-stimulated COX-2 expression. We further evaluated whether phosphorylation of JNK1/2 was involved in the SiNP-induced responses. We discovered that SiNPs stimulated phosphorylation of JNK1/2 in a time-dependent manner, which was attenuated by transfection with either Akt or JNK2 siRNA. However, SiNP-stimulated Akt phosphorylated was not mitigated by transfection with JNK2 siRNA (Figure 8D). Finally, SiNP-induced PGE_2_ production was decreased by SP600125 (Figure 8E). According to the above findings, the up-regulation of COX-2 expression and PGE_2_ release by SiNP challenge was mediated through Akt-dependent JNK1/2 activation in HPAEpiCs.

### 3.9. SiNPs Induce COX-2 Expression via FoxO1

This promoter region of COX-2 contains various putative transcriptional regulatory elements for the recognition of transcription factors such as FoxO1 or AP-1 [27,28,36]. Our previous study also demonstrated that FoxO1 participated in COX-2 induction [27]. Therefore, we investigated whether FoxO1 was involved in the SiNP-induced COX-2 expression in HPAEpiCs. For this purpose, we used a specific FoxO1 inhibitor AS1842856 and found that pretreatment of HPAEpiCs with AS1842856 significantly attenuated the SiNP-induced COX-2 protein expression (Figure 9A) and mRNA expression (Figure 9B). To ensure the role of FoxO1 in the SiNP-induced COX-2 expression, we adopted FoxO1 siRNA transfection which knocked down the FoxO1 protein level and also suppressed the SiNP-induced COX-2 expression (Figure 9C). Next, phosphorylation of FoxO1 was examined to clarify whether phosphorylation of FoxO1 was involved in the SiNP-induced response. As shown in Figure 9D, by siRNA transfection, we found that SiNP-induced FoxO1 phosphorylation was attenuated by transfection with FoxO1, p38α, or JNK2 siRNA. In addition, phosphorylation of p38 MAPK and JNK1/2 was not blocked by FoxO1 siRNA. Finally, pretreatment with AS1842856 significantly attenuated the SiNP-induced PGE_2_ release (Figure 9E). These results suggested that SiNP-induced COX-2 expression and PGE_2_ release are mediated through p38 MAPK- and JNK1/2-dependent activation of FoxO1 in HPAEpiCs.

### 3.10. SiNPs Induce COX-2 Expression via AP-1 in HPAEpiCs

It has been demonstrated that AP-1 plays a role in lung inflammatory responses and can be activated by MAPKs, thereby increasing its transcriptional activity to regulate the expression of cytokines and COX-2 in various cell types [28,37]. To determine whether SiNP-induced COX-2 expression mediated through transcription factor AP-1, we pretreated HPAEpiCs with an AP-1 inhibitor (Tanshinone IIA). The results in Figure 10A,B demonstrated that Tanshinone IIA significantly inhibited the SiNP-induced COX-2 protein expression in a concentration-dependent manner and COX-2 mRNA expression. Furthermore, we took advantage of siRNA transfection to ascertain the role of c-Jun (an AP-1 subunit) in the SiNP-induced COX2 expression. As shown in Figure 10C, the knockdown of c-Jun expression by c-Jun siRNA transfection significantly reduced COX-2 protein expression. In addition, transfection with either p38α or JNK2 siRNA also attenuated the SiNP-stimulated c-Jun phosphorylation, while transfection with c-Jun siRNA had no inhibitory effects on phosphorylation of p38 MAPK and JNK1/2 (Figure 10D). Finally, PGE_2_ secretion was attenuated by pretreatment with Tanshinone IIA in HPAEpiCs exposed to SiNPs (Figure 10E). Together, these results suggest that AP-1 mediated through p38 MAPK and JNK1/2 activation is involved in the SiNP-induced COX-2 expression and PGE_2_ release in HPAEpiCs.

## 4. Discussion

It has been established that prolonged inhalation of crystalline silica can lead to lung inflammation and the development of various diseases, including chronic bronchitis, lung fibrosis, and silicosis [38,39]. Moreover, the expression of COX-2 and PGs has been observed to be significantly induced in inflammatory pulmonary diseases such as COPD, acute lung injury, and asthma [11,12]. Previous research from our laboratory has also supported the involvement of COX-2 and PGE_2_ expression in pulmonary inflammatory responses triggered by external stimuli [14,15]. Given this context, our present study aims to investigate whether SiNPs can induce the up-regulation of the pro-inflammatory mediator COX-2 and the release of PGE_2_, and to elucidate the mechanisms through which SiNPs induce COX-2 expression in HPAEpiCs. Our results indicate that SiNP-induced COX-2 expression and PGE_2_ release are mediated through a cascade involving ROS/Pyk2/EGFR/PI3K/Akt/p38 MAPK and JNK1/2 pathways, ultimately leading to the activation of FoxO1 and AP-1 transcription factors in HPAEpiCs (Figure 11).

Production of ROS is initially recognized as a defense mechanism against pathogens in phagocytes during the respiratory burden [40]. However, it has been increasingly identified as having a crucial role in inflammation, cell damage, and even cell death directly or indirectly. According to previous studies, exposure to crystalline silica causes persistent lung inflammation and even fibrosis or apoptosis because of the sustained release of ROS in the alveolar space [31,41]. Therefore, the induction of oxidative stress is considered an important mechanism of SiNPs to cause lung insults [20]. We tried to eliminate ROS generation by free radical scavenger edaravone. Our results indicated that blockage of SiNP-induced ROS accumulation by edaravone markedly decreased COX-2 expression and PGE_2_ synthesis in HPAEpiCs. Kundu et al. revealed that SiNPs induce COX-2 expression through ROS-mediated activation of the Akt signaling pathway in HaCaT cells [23]. In addition, in human periodontal ligament cells, COX-2 expression is induced by titanium dioxide nanoparticles through ROS generation [42]. These findings are consistent with our present results indicating that ROS have a role in the SiNP-induced COX-2 expression. However, ROS are generated through several kinds of mechanisms, such as peroxisomes, xanthine oxidase, mitochondrial respiration, and the most common source NADPH oxidase [43]. We will explore these detailed mechanisms of ROS production stimulated by SiNPs in the future.

Pyk2 has been shown to play a key role in the development of pulmonary inflammation in mouse models and human neutrophils [32,44]. A previous report showed that SiO_2_-stimulated Pyk2 phosphorylation is involved in pulmonary fibroblast migration [45]. In this study, we also verified the involvement of Pyk2 in the SiNP-induced COX-2 expression in HPAEpiCs, which was inhibited by PF-431396 or transfection with Pyk2 siRNA. This response is mediated through Pyk2 phosphorylation which is inhibited by ROS scavenger edaravone or transfection with Pyk2 siRNA, implying that Pyk2 is downstream of ROS.

EGFR-activated signaling is involved in various cellular physiological and pathologic processes, especially in lung injury and inflammation [22,33,34]. SiNPs have been demonstrated cytotoxicity on breast cancer cells mediated through modulation of EGFR [46]. In this study, we found that pretreatment with EGFR inhibitor (AG1478) or transfection with EGFR siRNA lessened COX-2 expression in the SiNPs-treated HPAEpiCs, suggesting that EGFR also takes part in the SiNP-induced COX-2 expression. ROS unbalance intracellular phosphorylation and enhance the activity of intracellular PTKs [47]. It is indicated that non-receptor tyrosine kinases may be required to take part in the mechanism of ROS-induced EGFR transactivation [48]. Moreover, in vascular pathogenesis, the mechanisms by which ROS lead to transactivation of the EGFR and PDGFR and activation of non-receptor tyrosine kinases such as Pyk2 have been indicated [49]. Our data also found that ROS generation triggers Pyk2 activation. Additionally, our experiments revealed that transfection with either Pyk2 or EGFR siRNA blocked SiNP-induced EGFR phosphorylation, while transfection with EGFR siRNA had no effect on Akt phosphorylation. These results consistently clarify that SiNPs stimulate ROS/Pyk2-dependent EGFR activation, leading to COX-2 expression and PGE_2_ synthesis in HPAEpiCs.

The activation of EGFR has been shown to widely mediate the downstream effector pathways, including PI3K/Akt, STAT (signal transducer and activator of transcription), mTOR (mammalian target of rapamycin), and MAPKs. PI3Ks are activated by multiple cell-surface receptors such as receptor tyrosine kinases, G protein-coupled receptors (GPCRs), and other signaling complexes [50], which regulate cell movement, growth, survival, and differentiation. Many of these functions relate to the ability of PI3Ks to activate Akt in the PI3K/Akt pathway and its role in controlling the activation of FoxOs [51]. A previous study has uncovered that SiNPs stimulate ROS-related Akt activation to up-regulate COX-2 expression in human keratinocyte cells [23]. In the present study, we demonstrated that EGFR regulates the SiNP-induced COX-2 expression via PI3K/Akt pathway as essential downstream components which are blocked by EGFR inhibitor (AG1478) or its siRNA. Moreover, SiNP-stimulated Akt phosphorylation is inhibited by transfection with either EGFR or Akt siRNA, while transfection with Akt siRNA failed to change EGFR phosphorylation, indicating Akt is a downstream component of EGFR. Moreover, our data revealed that pretreatment with LY294002 or Akt inhibitor Ⅷ or transfection with their own siRNAs attenuated the SiNP-stimulated COX-2 expression and PGE_2_ synthesis. Thus, the present findings indicated that EGFR-dependent PI3K/Akt signaling pathway participates in the SiNP-induced COX-2 expression and PGE_2_ synthesis in HPAEpiCs.

MAPK cascades are major intracellular signalings that play an important role in various cellular processes including cell growth, differentiation, cell survival, cell death, and cellular stress and inflammatory responses [25]. The pro-inflammatory responses activated by nanoparticles on cells can be mediated through MAPK signaling [30]. Several studies have unveiled that SiNP-stimulated ROS-dependent MAPKs activation leading to vascular endothelial cell injury through apoptosis and autophagy [16,17]. In this study, we clarified that p38 MAPKs and JNK1/2 are involved in the SiNP-induced COX-2 expression and PGE_2_ secretion in HPAEpiCs by using the pharmacologic inhibitors of p38 MAPKs (p38 MAPK inhibitor VIII) and JNK1/2 (SP600125) or transfection with siRNA of p38α and JNK2 which reduced the SiNP-induced COX-2 expression and PGE_2_ synthesis, while the inhibitor of MEK1/2 (U0126) had no effect on these responses. Furthermore, SiNP stimulated the phosphorylation of p38 MAPK and JNK1/2 was attenuated by Akt siRNA. In addition, transfection with p38α or JNK2 siRNA attenuated the phosphorylation of p38 MAPK and JNK1/2 but had no change on Akt phosphorylation. These findings indicated that PI3K/Akt-dependent p38 MAPK and JNK1/2 activation participate in the SiNP-stimulated COX-2 expression and PGE_2_ synthesis in HPAEpiCs. These results are also consistent with several reports showing that induction of COX-2 and phosphorylation of p38 MAPK by SiNPs was demonstrated in A549 cells [7]. MAPK signaling pathways are up-regulated by SiNPs to induce gene transcription including COX-2 in several kinds of cells [30]. Thus, our findings suggested that SiNP-stimulated COX-2 expression and PGE_2_ synthesis are mediated through Akt-dependent p38 MAPK and JNK1/2 activation in HPAEpiCs.

The activity of FoxOs is tightly regulated by a variety of post-translational modifications, which can either activate or inhibit FoxO activity that is involved in several pathological and physiologic processes including proliferation, apoptosis, autophagy, metabolism, inflammation, and resistance to oxidative stress in several types of cells [52]. PI3K/Akt signaling pathway is a major regulator of FoxO activity. FoxO1 is a well-known member of the FoxO family. As in our previous studies, cytokine-induced up-regulation of COX-2 and PGE_2_ are mediated through p38 MAPK and J

NK1/2-dependent FoxO1 activation in fibroblasts [27]. The present results are the first time to clarify the involvement of FoxO1 in the SiNP-induced COX-2 and PGE_2_ expression by a specific FoxO1 inhibitor, AS1842856, and FoxO1 siRNA transfection. Moreover, we verified the FoxO1 activity was regulated by p38 MAPK and JNK1/2 activities, due to the transfection of cells with FoxO1, p38 MAPK, or JNK2, siRNA attenuated FoxO1 activity induced by SiNPs in HPAEpiCs. In addition, p38 MAPK and JNK1/2 phosphorylation were not changed by transfection with FoxO1 siRNA.

The promoter region of COX-2 contains several potential transcription regulatory elements such as AP-1, NF-κB, CRE, and Sp1, with a little bit of difference dependent on the cell types [36]. c-Fos and c-Jun form a heterodimer, creating the AP-1 complex, which plays a vital role in regulating gene expression in response to extracellular signals [36,37]. This complex binds to specific sites in the promoter and enhancer regions of target genes. By doing so, it enables the conversion of extracellular signals into changes in gene expression. Among the AP-1 components, c-Fos and c-Jun have been extensively studied. They possess several homologous domains, including adjacent basic and leucine zipper motifs. These domains are essential for DNA binding and dimerization, respectively. While c-Jun has the ability to homodimerize, it shows a preference for heterodimerization with partners like c-Fos [37]. AP-1 activity is regulated by a broad range of physiological and pathological stimuli, including cytokines, growth factors, stress signals, and infections, which activate the MAPK cascades leading to the transcription and phosphorylation of c-Fos and c-Jun and enhanced transcriptional activity [37]. The activation of JNK/AP-1 may lead to the induction of pro-inflammatory and pro-apoptotic gene expression under SiNP exposure [30]. Moreover, the COX-2 gene was found to be up-regulated upon SiNP treatment due to AP-1-mediated gene transcription [30]. We applied an AP-1 transcription factor inhibitor Tanshinone IIA and c-Jun siRNA to demonstrate that AP-1 is required for COX-2 expression induced by SiNPs in HPAEpiCs. In addition, we analyzed the role of MAPKs in the SiNP-induced AP-1 activation by using transfection with p38α or JNK2 siRNA. Transfection of cells with either c-Jun, p38 MAPK, or JNK2 siRNA suppressed c-Jun phosphorylation. On the other hand, we found that p38 MAPK and JNK1/2 were involved in SiNPs-mediated c-Jun, ATF2, and JunD, but not JunB, phosphorylation in these cells (see Appendix A). Thus, our results proved that SiNPs induce COX-2 expression and PGE_2_ levels via MAPK-dependent FoxO1 and AP-1 (c-Jun, ATF2, and JunD) activation in HPAEpiCs.

## 5. Conclusions

The present study concludes that SiNP-induced up-regulation of COX-2 and PGE_2_ synthesis is, at least partially, mediated through ROS accumulation, resulting in the transactivation of EGFR by Pyk2. Subsequently, the PI3K/Akt pathway is activated by EGFR, leading to concurrent p38 MAPK- and JNK1/2-dependent activation of AP-1 and FoxO1. These transcription factors bind to the AP-1 binding site and FoxO1 response element (FRE) on the COX-2 promoter, thereby increasing the expression of COX-2 and PGE_2_ synthesis induced by SiNPs. The COX-2/PGE_2_ axis appears to be involved in the inflammatory responses in HPAEpiCs. It is important to note that one limitation of the present report is the absence of in vivo data. Future research should explore animal models to further investigate these mechanisms, which could offer valuable insights into the development of anti-inflammatory therapeutics for treating pulmonary inflammation.

## Figures and Tables

**Figure 1 biomedicines-11-02628-f001:**
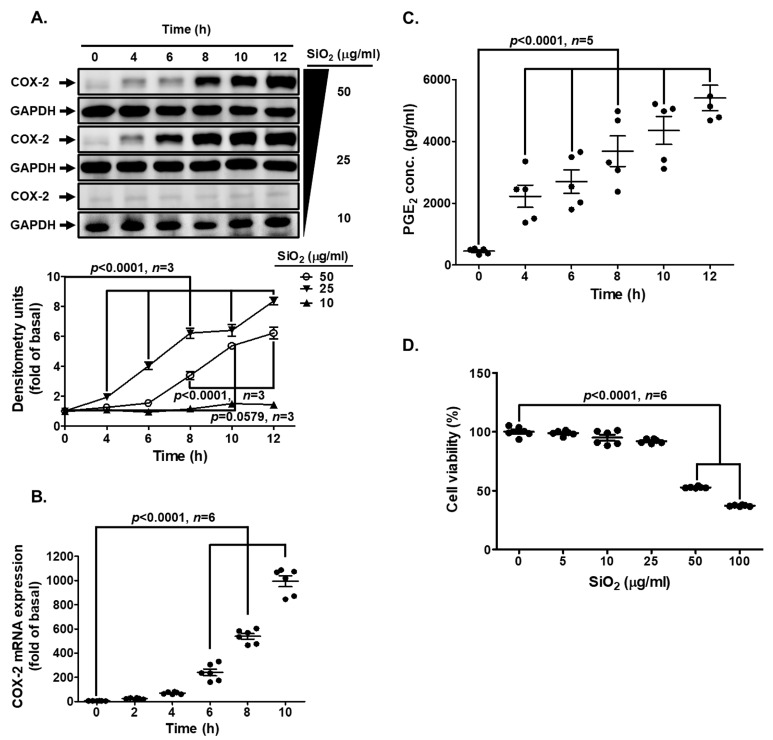
**SiNPs induce COX-2 protein and mRNA expression, as well as PGE_2_ production in HPAEpiCs.** (**A**) HPAEpiCs were incubated with different concentrations (10, 25, and 50 μg/mL) of SiNPs for the indicated time intervals (0, 4, 6, 8, 10, and 12 h). The levels of COX-2 and GAPDH protein were determined by Western blot. (**B**) HPAEpiCs were treated with 25 μg/mL SiNPs for the indicated time intervals (0, 2, 4, 6, 8, 10 h). The COX-2 and GAPDH mRNA levels were determined by quantitative real-time PCR. (**C**) The culture media from (**A**), treated with 25 μg/mL SiNPs, were collected to determine the levels of PGE_2_ using a PGE_2_ ELISA kit. (**D**) Cells were incubated with different concentrations (5, 10, 25, 50, and 100 μg/mL) of SiNPs for 12 h. Cell viability was determined using the Cell Counting Kit 8. Data are presented as mean ± S.E.M. from at least three independent experiments as indicated on each panel.

**Figure 2 biomedicines-11-02628-f002:**
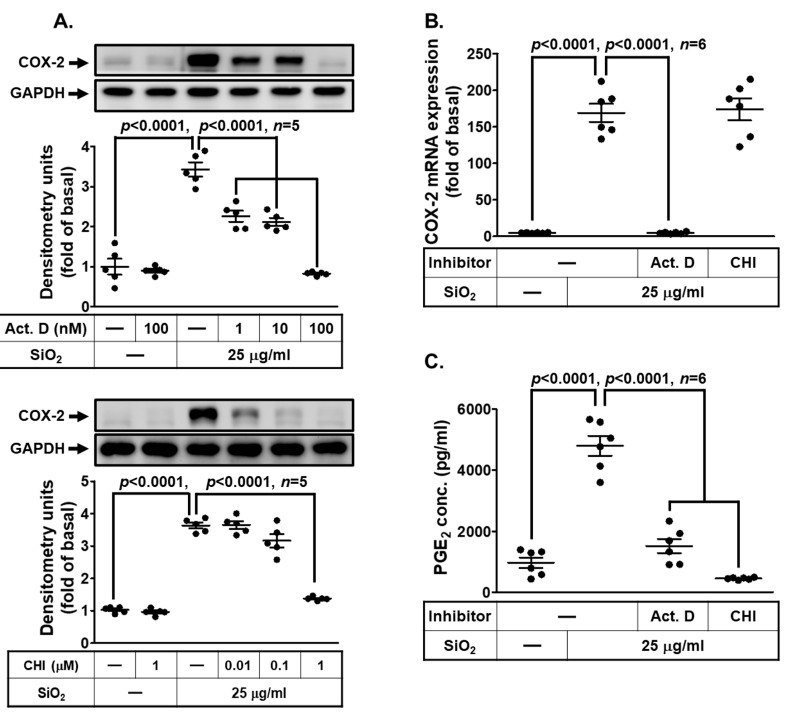
**SiNP-induced COX-2 expression and PGE_2_ production require ongoing transcription and translation in HPAEpiCs.** (**A**) HPAEpiCs were pretreated with various concentrations of Act. D (1, 10, and 100 nM) or CHI (0.01, 0.1, and 1 μM) for 1 h, respectively, and then incubated with 25 μg/mL SiNPs for 12 h. COX-2 and GAPDH protein expression were determined by Western blot. (**B**) HPAEpiCs were pretreated with Act. D (100 nM) or CHI (1 μM) for 1 h, and then incubated with 25 μg/mL SiNPs for 6 h. The levels of COX-2 and GAPDH mRNA were analyzed by quantitative real-time PCR. (**C**) The culture media from (**A**) were collected to determine the levels of PGE_2_ synthesis using a PGE_2_ ELISA kit. Data are presented as mean ± S.E.M. from at least three independent experiments as indicated on each panel.

**Figure 3 biomedicines-11-02628-f003:**
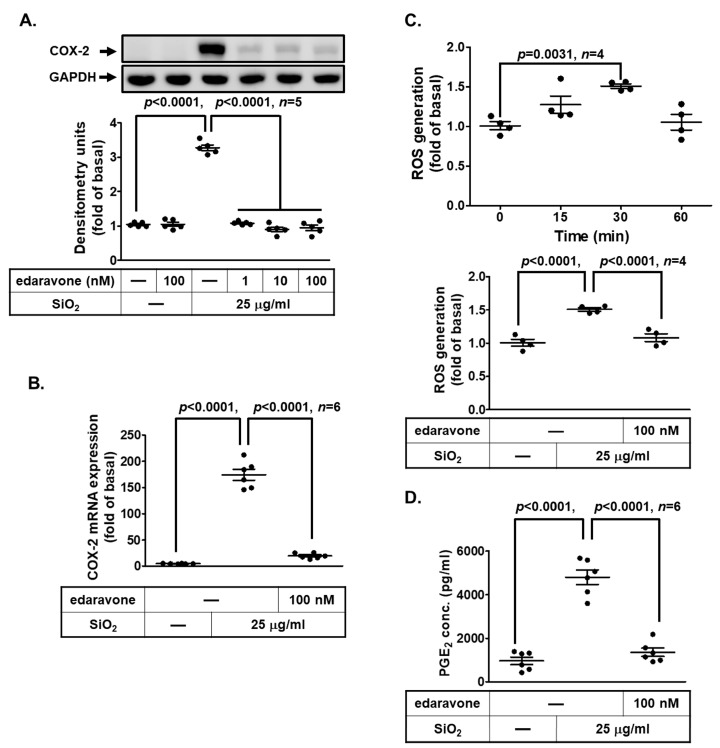
**ROS generation induced by SiNPs mediates COX-2 expression and PGE_2_ production in HPAEpiCs.** (**A**) HPAEpiCs were pretreated with various concentrations of edaravone (1, 10, and 100 nM) for 1 h, and then incubated with 25 μg/mL SiNPs for 12 h. COX-2 and GAPDH protein expression were determined by Western blot. (**B**) Cells were pretreated with edaravone (0.1 μM) for 1 h, and then incubated with 25 μg/mL SiNPs for 6 h. The COX-2 and GAPDH mRNA levels were determined by quantitative real-time PCR. (**C**) Cells were stimulated with 25 μg/mL SiNPs for 15, 30, and 60 min (upper panel) and were pretreated without or with edaravone (100 nM) for 1 h before exposure to 25 μg/mL SiNPs for 30 min (lower panel), followed by incubation with H_2_DCF-DA (5 μM) for 30 min. The fluorescence intensity of cells was measured using a fluorescent microplate reader. (**D**) The culture media from (**A**) were collected to determine the levels of PGE_2_ synthesis using a PGE_2_ ELISA kit. Data are presented as mean ± S.E.M. from at least three independent experiments as indicated on each panel.

**Figure 4 biomedicines-11-02628-f004:**
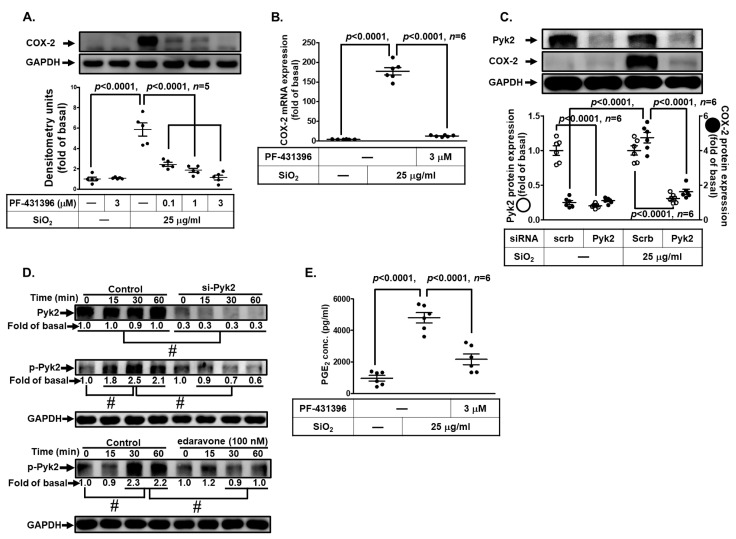
**SiNPs induced COX-2 expression and PGE_2_ production via Pyk2 activation in HPAEpiCs.** (**A**) HPAEpiCs were pretreated with various concentrations of PF-431396 (0.1, 1, and 3 μM) for 1 h, and then incubated with 25 μg/mL SiNPs for 12 h. The protein expression of COX-2 and GAPDH was determined by Western blot. (**B**) HPAEpiCs were pretreated with PF-431396 (3 μM) for 1 h, and then incubated with 25 μg/mL SiNPs for 6 h. The levels of COX-2 and GAPDH mRNA were analyzed by quantitative real-time PCR. (**C**) The cells were transfected with scrambled or Pyk2 siRNA, and then incubated with 25 μg/mL SiNPs for 12 h. The protein levels of COX-2, Pyk2, and GAPDH were analyzed by Western blot. (**D**) The cells were transfected with scrambled siRNA as a control and Pyk2 siRNA, and pretreated without or with 100 nM edaravone for 1 h. They were then incubated with 25 μg/mL SiNPs for the indicated time intervals (15, 30, and 60 min). The protein levels of phospho-Pyk2, total Pyk2, and GAPDH were analyzed by Western blot. (**E**) The culture media from (**A**) were collected to determine the levels of PGE_2_ synthesis using a PGE_2_ ELISA kit. Data are presented as mean ± S.E.M. from at least three independent experiments as indicated on each panel. # *p* < 0.01, as compared between the two indicated groups.

**Figure 5 biomedicines-11-02628-f005:**
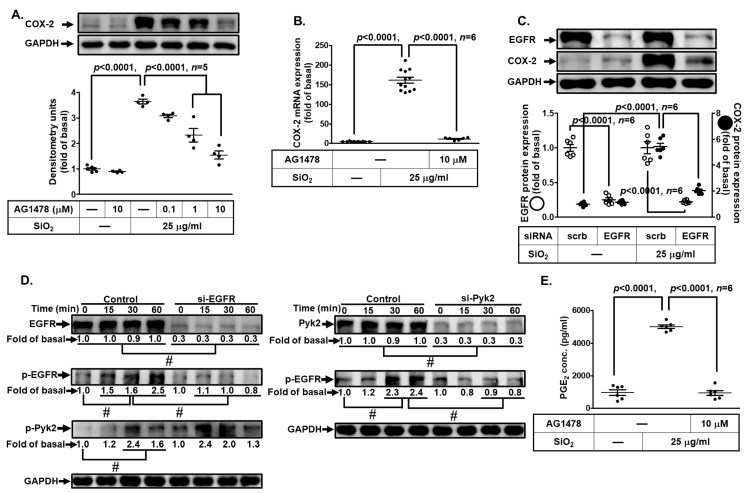
**EGFR is required for SiNP-induced COX-2 expression and PGE_2_ production in HPAEpiCs.** (**A**) HPAEpiCs were pretreated with various concentrations of AG1478 (0.1, 1, and 10 μM) for 1 h, and then incubated with 25 μg/mL SiNPs for 12 h. The protein expression of COX-2 and GAPDH was determined by Western blot. (**B**) HPAEpiCs were pretreated with AG1478 (10 μM) for 1 h, and then incubated with 25 μg/mL SiNPs for 6 h. The levels of COX-2 and GAPDH mRNA were analyzed by quantitative real-time PCR. (**C**) The cells were transfected with scrambled or EGFR siRNA, and then incubated with 25 μg/mL SiNPs for 12 h. The protein levels of COX-2, EGFR, and GAPDH were analyzed by Western blot. (**D**) The cells were transfected with scrambled siRNA as a control and EGFR or Pyk2 siRNA, and then incubated with 25 μg/mL SiNPs for the indicated time intervals (15, 30, and 60 min). The protein levels of phospho-Pyk2, total Pyk2, phospho-EGFR, total EGFR, and GAPDH were analyzed by Western blot. (**E**) The culture media from (**A**) were collected to determine the levels of PGE_2_ synthesis using a PGE_2_ ELISA kit. Data are presented as mean ± S.E.M. from at least three independent experiments as indicated on each panel. # *p* < 0.01, as compared between the two indicated groups.

**Figure 6 biomedicines-11-02628-f006:**
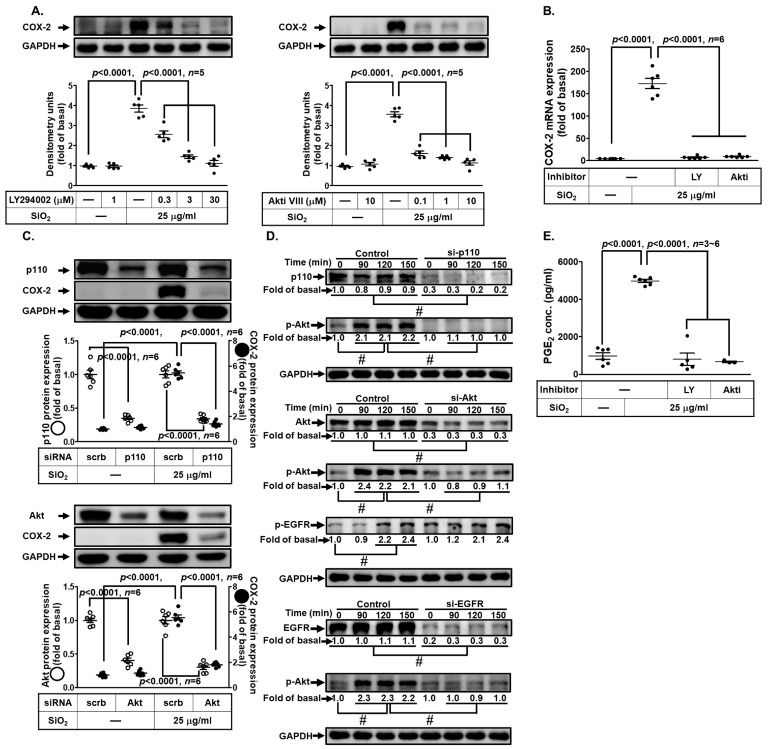
**PI3K/Akt signaling involvement in SiNP-induced COX-2 expression and PGE_2_ secretion in HPAEpiCs.** (**A**) HPAEpiCs were pretreated with various concentrations of LY294002 (0.3, 3, and 30 μM) or Akt inhibitor VIII (0.1, 1, and 10 μM) for 1 h, and then incubated with 25 μg/mL SiNPs for 12 h. The protein expression of COX-2 and GAPDH was determined by Western blot. (**B**) Cells were pretreated with LY294002 (30 μM) or Akt inhibitor VIII (10 μM) for 1 h, and then incubated with 25 μg/mL SiNPs for 6 h. The levels of COX-2 and GAPDH mRNA were determined by quantitative real-time PCR. (**C**) The cells were transfected with scrambled, p110, or Akt siRNA, and then incubated with 25 μg/mL SiNPs for 12 h. The protein levels of COX-2, p110, Akt, and GAPDH were analyzed by Western blot. (**D**) The cells were transfected with scrambled siRNA as a control, and p110, Akt, or EGFR siRNA, and then incubated with 25 μg/mL SiNPs for the indicated time intervals (90, 120, and 150 min). The protein levels of phospho-Akt, total Akt, phospho-EGFR, total EGFR, total p110, and GAPDH were analyzed by Western blot. (**E**) The culture media from (**A**) were collected to determine the levels of PGE_2_ synthesis using a PGE_2_ ELISA kit. Data are presented as mean ± S.E.M. from at least three independent experiments as indicated on each panel. # *p* < 0.01, as compared between the two indicated groups.

**Figure 7 biomedicines-11-02628-f007:**
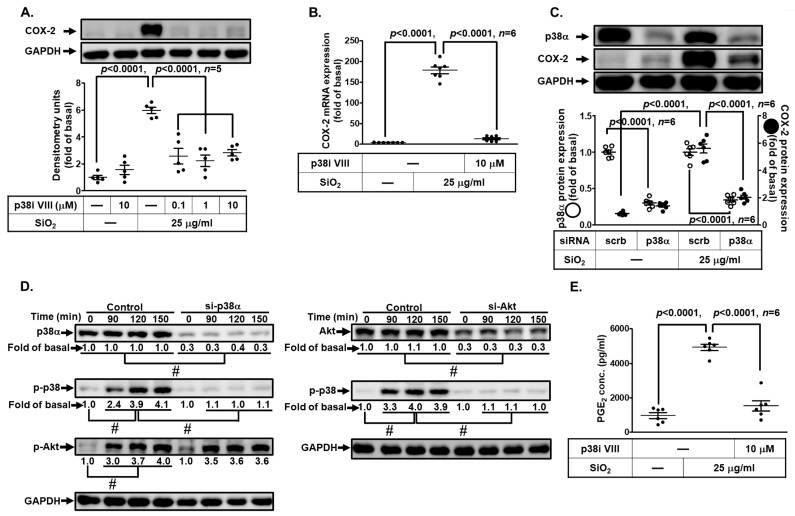
**p38 MAPK involvement in SiNP-induced COX-2 expression and PGE_2_ production in HPAEpiCs.** (**A**) HPAEpiCs were pretreated with various concentrations of p38 MAPK inhibitor VIII (0.1, 1, and 10 μM) for 1 h, and then incubated with 25 μg/mL SiNPs for 12 h. The protein expression of COX-2 and GAPDH was determined by Western blot. (**B**) HPAEpiCs were pretreated with p38 MAPK inhibitor VIII (10 μM) for 1 h, and then incubated with 25 μg/mL SiNPs for 6 h. The levels of COX-2 and GAPDH mRNA were analyzed by quantitative real-time PCR. (**C**) The cells were transfected with scrambled or p38 siRNA, and then incubated with 25 μg/mL SiNPs for 12 h. The protein levels of COX-2, p38, and GAPDH were analyzed by Western blot. (**D**) The cells were transfected with scrambled siRNA as control, and p38 siRNA or Akt siRNA, and then incubated with 25 μg/mL SiNPs for the indicated time intervals (90, 120, and 150 min). The protein levels of phospho-Akt, total Akt, phospho-p38, total p38, and GAPDH were analyzed by Western blot. (**E**) The culture media from (**A**) were collected to determine the levels of PGE_2_ synthesis using a PGE_2_ ELISA kit. Data are presented as mean ± S.E.M. from at least three independent experiments as indicated on each panel. # *p* < 0.01, as compared between the two indicated groups.

**Figure 8 biomedicines-11-02628-f008:**
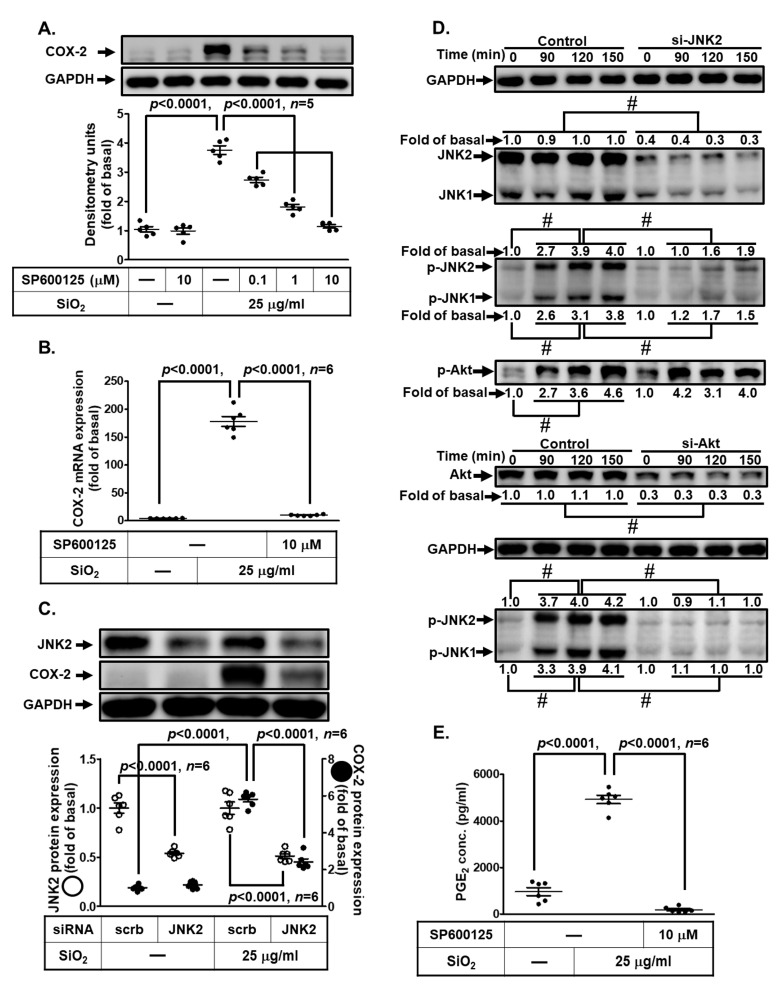
**JNK1/2 contributes to SiNP-induced COX-2 expression and PGE_2_ production in HPAEpiCs.** (**A**) HPAEpiCs were pretreated with various concentrations of SP600125 (0.1, 1, and 10 μM) for 1 h, and then incubated with 25 μg/mL SiNPs for 12 h. The protein expression of COX-2 and GAPDH was determined by Western blot. (**B**) HPAEpiCs were pretreated with SP600125 (10 μM) for 1 h, and then incubated with 25 μg/mL SiNPs for 6 h. The levels of COX-2 and GAPDH mRNA were analyzed by quantitative real-time PCR. (**C**) The cells were transfected with scrambled or JNK2 siRNA, and then incubated with 25 μg/mL SiNPs for 12 h. The protein levels of COX-2, JNK2, and GAPDH were analyzed by Western blot. (**D**) The cells were transfected with scrambled siRNA as control, and JNK2 or Akt siRNA, and then incubated with 25 μg/mL SiNPs for the indicated time intervals (90, 120, and 150 min). The protein levels of phospho-Akt, total Akt, phospho-JNK1/2, total JNK1/2, and GAPDH were analyzed by Western blot. (**E**) The culture media from (**A**) were collected to determine the levels of PGE_2_ synthesis using a PGE_2_ ELISA kit. Data are presented as mean ± S.E.M. from at least three independent experiments as indicated on each panel. # *p* < 0.01, as compared between the two indicated groups.

**Figure 9 biomedicines-11-02628-f009:**
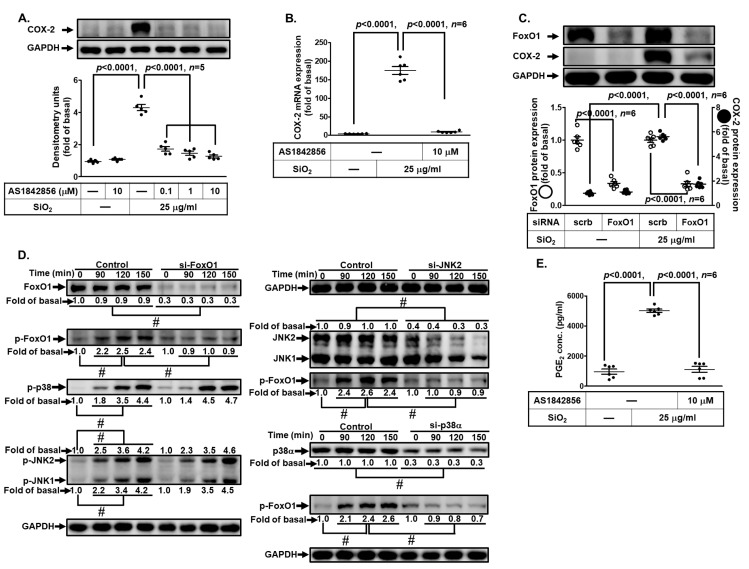
**Involvement of FoxO1 in SiNP-induced COX-2 expression and PGE_2_ production in HPAEpiCs.** (**A**) HPAEpiCs were pretreated with various concentrations of AS1842856 (0.1, 1, and 10 μM) for 1 h, and then incubated with 25 μg/mL SiNPs for 12 h. The protein expression of COX-2 and GAPDH was determined by Western blot. (**B**) Cells were pretreated with AS1842856 (10 μM) for 1 h, and then incubated with 25 μg/mL SiNPs for 6 h. The levels of COX-2 and GAPDH mRNA were determined by quantitative real-time PCR. (**C**) The cells were transfected with scrambled or FoxO1 siRNA, and then incubated with 25 μg/mL SiNPs for 12 h. The protein levels of COX-2, FoxO1, and GAPDH were analyzed by Western blot. (**D**) The cells were transfected with scrambled siRNA as control, and FoxO1, JNK2, or p38 siRNA, and then incubated with 25 μg/mL SiNPs for the indicated time intervals (90, 120, and 150 min). The protein levels of phospho-FoxO1, total FoxO1, phospho-JNK1/2, total JNK1/2, phospho-p38, total p38, and GAPDH were analyzed by Western blot. (**E**) The culture media from (**A**) were collected to determine the levels of PGE_2_ synthesis using a PGE_2_ ELISA kit. Data are presented as mean ± S.E.M. from at least three independent experiments as indicated on each panel. # *p* < 0.01, as compared between the two indicated groups.

**Figure 10 biomedicines-11-02628-f010:**
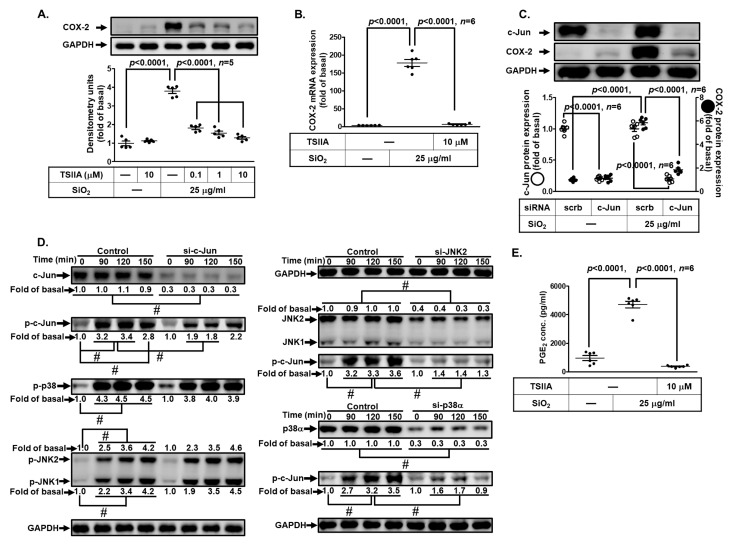
**c-Jun/AP-1 regulates COX-2 expression and PGE_2_ production induced by SiNPs in HPAEpiCs.** (**A**) HPAEpiCs were pretreated with various concentrations of Tanshinone IIA (0.1, 1, and 10 μM) for 1 h, and then incubated with 25 μg/mL SiNPs for 12 h. The protein expression of COX-2 and GAPDH was determined by Western blot. (**B**) Cells were pretreated with Tanshinone IIA (10 μM) for 1 h, and then incubated with 25 μg/mL SiNPs for 6 h. The levels of COX-2 and GAPDH mRNA were determined by quantitative real-time PCR. (**C**) The cells were transfected with scrambled or c-Jun siRNA, and then incubated with 25 μg/mL SiNPs for 12 h. The protein levels of COX-2, c-Jun, and GAPDH were analyzed by Western blot. (**D**) The cells were transfected with scrambled siRNA as control, and c-Jun siRNA, JNK2 siRNA, or p38 siRNA, and then incubated with 25 μg/mL SiNPs for the indicated time intervals (90, 120, and 150 min). The protein levels of phospho-c-Jun, total c-Jun, phospho-JNK1/2, total JNK1/2, phospho-p38, total p38, and GAPDH were analyzed by Western blot. (**E**) The culture media from (**A**) were collected to determine the levels of PGE_2_ synthesis using a PGE_2_ ELISA kit. Data are presented as mean ± S.E.M. from at least three independent experiments as indicated on each panel. # *p* < 0.01, as compared between the two indicated groups.

**Figure 11 biomedicines-11-02628-f011:**
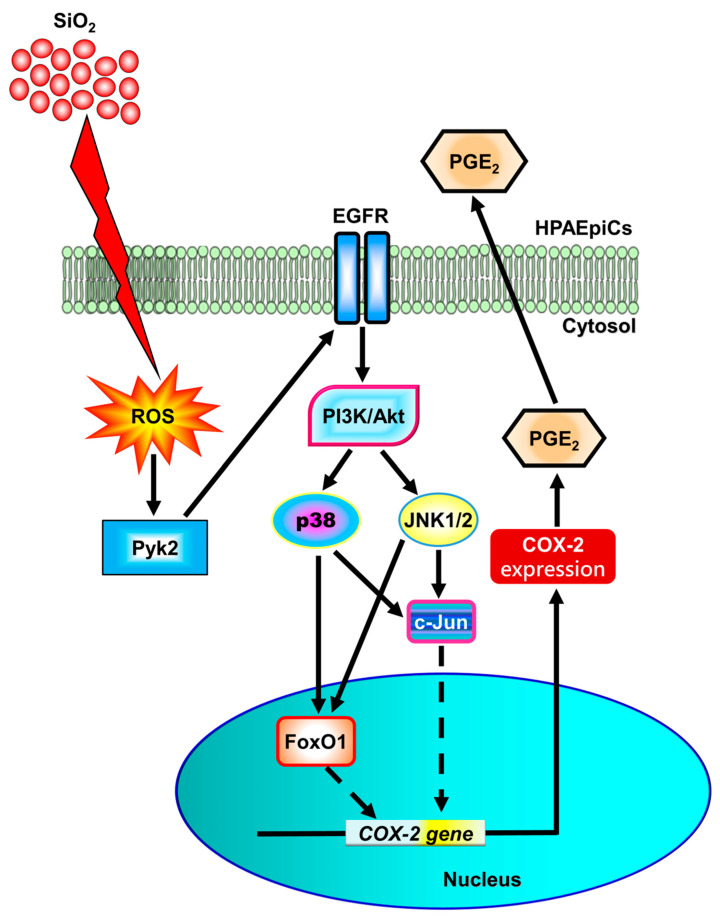
**Schematic signaling pathways contributing to SiNP-induced COX-2 expression and PGE_2_ production in HPAEpiCs.** SiNP-induced up-regulation of COX-2 and synthesis of PGE_2_ are, at least in part, mediated through ROS accumulation, which leads to the transactivation of EGFR by Pyk2. Subsequently, the activation of the PI3K/Akt pathway is triggered by EGFR, resulting in the activation and translocation of c-Jun and FoxO1 from the cytosol to the nucleus in a p38 MAPK- and JNK1/2-dependent manner. The activation of AP-1 and FoxO1 contributes to the increased levels of COX-2 and PGE_2_ induced by SiNPs. Ultimately, the COX-2/PGE_2_ axis may promote inflammatory responses in HPAEpiCs.

## Data Availability

The data presented in this study are available on request from the corresponding author.

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
