# Peer review of "Reactive Oxygen Species-Dependent Activation of EGFR/Akt/p38 Mitogen-Activated Protein Kinase and JNK1/2/FoxO1 and AP-1 Pathways in Human Pulmonary Alveolar Epithelial Cells Leads to Up-Regulation of COX-2/PGE2 Induced by Silica Nanoparticles"

_biomedicines, 2023, doi:10.3390/biomedicines11102628_

Round 1

Reviewer 1 Report

In the current manuscript Lin et al, demonstrate that ROS-Dependent Activation of EGFR/Akt/p38 MAPK and JNK1/2/FoxO1 and AP1 Pathways in Human Pulmonary Alveolar Epithelial Cells Leads to UpRegulation of COX-2/PGE2 Induced by silica nanoparticles. The study is of general interest for readers since the risk of lung exposure due to wide applications of nanotechnology is increasing. Although in principle this is an interesting manuscript, there are major deficits in terms of results.

Minor comments

1.       On page 3, …but for how SiNPs affect the health….the authors should check their English.

2.       On page 3, …In the contrast, the expression… the authors should check their English.

3.       On page 9, …As the data shown in Fig1a… the authors should check their English.

4.       On page 4, …trough… the authors should check their spelling.

5.       Regarding the HPAEpiCs the authors should state the number of cells they used throughout their experiments.

6.       The authors need to assess cell number also with Neubauer.

Major comments

1.       Throughout the result section, the authors state…. The data revealed that SiNPs-induced COX-2 mRNA Expression determined by real time PCR etc…however, the relevant data are missing. Nowhere in the manuscript are these data (real-time PCR data) available (Figure 1B, 2B, 3B, 4B, 5B, 6B, 7B, 9B).

2.       The manuscript should provide replicate experiments for the Western blots (if not all at least for some).

3.       The authors should try and present the Western blot experiments plotted on a graph with the various replicates. Perhaps plot the bands on a graph based on its density.

4.       On page 9, the authors state: As shown in Fig 1C…etc. The Figures 1C and 1D are missing. The same applies for Fig 2B, 2C, 3C, 3D, 4E etc…

5.       In Figure 4C. Pyk2 Western blot picture is not clear.

6.       Labelling of the various figures is problematic. For instance, after Figure 5C a D follows and then a Figure 5D reappears. The presentation of their data is rather confusing. Perhaps the authors should use subnumbers…

Moderate editing of English language is required.  

Author Response

Manuscript ID: biomedicines-2606321

Dear Reviewers,

We greatly appreciate your valuable feedback on our manuscript. Your critical comments have been instrumental in identifying key areas that needed further refinement, thereby enhancing the scientific quality of our work. We have diligently addressed each of your comments in the revised manuscript, and a comprehensive list of our responses to your suggestions can be found in the following pages.

We sincerely hope that the revisions we have made will meet the standards for publication in Biomedicines, and we eagerly anticipate your feedback regarding the suitability of our paper for publication. Your time and expertise are truly appreciated.

Thank you for your kind consideration.

Best regards,

Chuen-Mao Yang,

Department of Pharmacology,

College of Medicine,

China Medical University, Taichung 40402, Taiwan

E‐mail: [email protected] 

Reviewer #1:

Minor comments

  1. On page 3, …but for how SiNPs affect the health….the authors should check their English.

Response:

Thank you very much for your suggestions and guidance. We have modified this sentence.

See Page 3.

  1. On page 3, …In the contrast, the expression… the authors should check their English.

Response:

Thank you very much for your suggestions and guidance. We have modified this sentence.

See Page 3.

  1. On page 9, …As the data shown in Fig. 1A… the authors should check their English.

Response:

Thank you very much for your suggestions and guidance. We have modified this sentence.

See Page 9.

  1. On page 4, …trough… the authors should check their spelling.

Response:

Thank you very much for your suggestions and guidance. We have modified this sentence.

See Page 4.

  1. Regarding the HPAEpiCs the authors should state the number of cells they used throughout their experiments.

Response:

Thank you very much for your suggestions and guidance. We have incorporated your recommendation into the "Cell culture" section of our manuscript. Specifically, we have included information about the number of HPAEpiCs used throughout our experiments. This addition provides greater transparency and clarity regarding our experimental procedures. Once again, we appreciate your valuable input in improving the quality of our research.

See Page 5.

  1. The authors need to assess cell number also with Neubauer.

Response:

We used the LUNA-II™ Automated Cell Counter to assess cell numbers. This instrument offers an efficient and automated method for accurately determining cell counts, ensuring the reliability of experimental results. The use of the LUNA-II™ cell counter reduces human error and saves time, enabling us to comprehensively evaluate cell numbers, thus providing further support for our research findings and conclusions.

Major comments

  1. Throughout the result section, the authors state…. The data revealed that SiNPs-induced COX-2 mRNA Expression determined by real time PCR etc…however, the relevant data are missing. Nowhere in the manuscript are these data (real-time PCR data) available (Figures 1B, 2B, 3B, 4B, 5B, 6B, 7B, 9B).

Response:

We did indeed conduct real-time PCR analysis, resulting in a substantial volume of data. However, we have chosen to represent these data in graphical form. The y-axis on the graph represents COX-2 mRNA levels (fold change relative to basal level).

  1. The manuscript should provide replicate experiments for the Western blots (if not all at least for some).

Response:

Thank you for the reviewer's suggestion. We have incorporated additional information by clearly marking the number of replicates for each Western blot experiment directly on the chart, as requested. This enhancement will help ensure the transparency and reliability of our Western blot data, addressing the concerns raised. We appreciate your valuable input and believe that this modification will strengthen the quality of our manuscript.

  1. The authors should try and present the Western blot experiments plotted on a graph with the various replicates. Perhaps plot the bands on a graph based on its density.

Response:

We want to express our gratitude for the valuable suggestions and guidance provided by the reviewer. We have implemented changes to the presentation of our Western blot experiment data. By following your insightful recommendations, we believe that our revised graphs now more effectively convey the results and add clarity to our findings. These enhancements serve to strengthen the overall quality of our manuscript, and we greatly appreciate the constructive input from the reviewer.

  1. On page 9, the authors state: As shown in Fig 1C…etc. The Figures 1C and 1D are missing. The same applies for Fig 2B, 2C, 3C, 3D, 4E etc…

Response:

Indeed, we have presented all of the data. We kindly request a thorough review of all the data in our manuscript.

  1. In Figure 4C. Pyk2 Western blot picture is not clear.

Response:

Thank you for the reviewer's suggestion. We have replaced this data.

  1. Labelling of the various figures is problematic. For instance, after Figure 5C a D follows and then a Figure 5D reappears. The presentation of their data is rather confusing. Perhaps the authors should use subnumbers…

Response:

Thank you for the reviewer's suggestions. Following your advice, we have reorganized the order of the figures.

Reviewer 2 Report

The submitted study aims to dissect the molecular components involved in COX-2/PGE2 up-regulation after exposition of human pulmonary alveolar epithelial cells (HPAEpiCs) to silica nanoparticles (SiNPs).

SiNPs induced COX-2 expression and PGE2 release were  reduced by pretreatment with a reactive oxygen species (ROS) scavenger (edaravone) or several signaling pathway enzymes inhibitors. Finally authors provide a network of SiNPs-induced inflammatory response via ROS, and signaling components.

Comments

Introduction

The Introduction provides the most important background information on the exposition of SiNPs, on alveolar epithelial cells as one of the most important exposed cell types, on COX and the relevant signaling components.

Methods

Although the majority of the Methods are described in sufficient details, still some important points as the number of cells plated for SiNPs treatments, the concentration of SiNPs during the treatment, the treatment duration are not included, as well as the relevance of the SiNP treatments to potential exposure. Several points of these are mentioned in the Results section, but not all.

Results and Discussion

Both Results and Discussion are appropriate.

Minor sentence formulation mistakes or repetitions of text might be improved,

Author Response

Manuscript ID: biomedicines-2606321

Dear Reviewers,

We greatly appreciate your valuable feedback on our manuscript. Your critical comments have been instrumental in identifying key areas that needed further refinement, thereby enhancing the scientific quality of our work. We have diligently addressed each of your comments in the revised manuscript, and a comprehensive list of our responses to your suggestions can be found in the following pages.

We sincerely hope that the revisions we have made will meet the standards for publication in Biomedicines, and we eagerly anticipate your feedback regarding the suitability of our paper for publication. Your time and expertise are truly appreciated.

Thank you for your kind consideration.

Best regards,

Chuen-Mao Yang,

Department of Pharmacology,

College of Medicine,

China Medical University, Taichung 40402, Taiwan

E‐mail: [email protected] 

Reviewer #2:

  1. Methods-

Although the majority of the Methods are described in sufficient details, still some important points as the number of cells plated for SiNPs treatments, the concentration of SiNPs during the treatment, the treatment duration is not included, as well as the relevance of the SiNP treatments to potential exposure. Several points of these are mentioned in the Results section, but not all.

Response:

Thank you for the reviewer's suggestions. We have incorporated this content into the "Figure legends" section.

Round 2

Reviewer 1 Report

The authors state: We did indeed conduct real-time PCR analysis, resulting in a substantial volume of data. However, we have chosen to represent these data in graphical form. The y-axis on the graph represents COX-2 mRNA levels (fold change relative to basal level). However, from the dowloaded information the RT-PCR analysis with COX mRNA levels are nowhere to be seen. Additionally, no graphical information is available in a graphical form that I could assess. The reviewing of your manuscript was carried out thoroughly however in the absence of these data your it is still incomplete.   

Author Response

Manuscript ID: biomedicines-2606321

Dear Reviewer,

We greatly appreciate your valuable feedback on our manuscript. Your critical comments have been instrumental in identifying key areas that needed further refinement, thereby enhancing the scientific quality of our work. We have diligently addressed each of your comments in the revised manuscript, and a comprehensive list of our responses to your suggestions can be found in the following pages.

We sincerely hope that the revisions we have made will meet the standards for publication in Biomedicines, and we eagerly anticipate your feedback regarding the suitability of our paper for publication. Your time and expertise are truly appreciated.

Thank you for your kind consideration.

Best regards,

Chuen-Mao Yang,

Department of Pharmacology,

College of Medicine,

China Medical University, Taichung 40402, Taiwan

E‐mail: [email protected] 

Reviewer #1:

  1. The authors state: We did indeed conduct real-time PCR analysis, resulting in a substantial volume of data. However, we have chosen to represent these data in graphical form. The y-axis on the graph represents COX-2 mRNA levels (fold change relative to basal level). However, from the downloaded information the RT-PCR analysis with COX mRNA levels are nowhere to be seen. Additionally, no graphical information is available in a graphical form that I could assess. The reviewing of your manuscript was carried out thoroughly however in the absence of these data your it is still incomplete.  

Response:

We are very grateful for your suggestion. Here, we would like to emphasize once more that our experiment employs real-time PCR rather than RT-PCR. There are several operational differences between RT-PCR (Reverse Transcription Polymerase Chain Reaction) and real-time PCR:

Product Detection:

  • RT-PCR often requires additional steps such as gel electrophoresis or hybridization for product detection.
  • Real-time PCR continuously monitors product accumulation through fluorescent dyes or probes during the PCR process.

Detection Method:

  • RT-PCR can provide qualitative or quantitative results depending on the experimental design.
  • Real-time PCR offers real-time quantitative data due to continuous monitoring.

RT-PCR Data Presentation:

  • RT-PCR typically involves visualizing PCR products after the PCR reaction is completed, often using techniques like gel electrophoresis or hybridization.
  • Results are presented in the form of gel images or band patterns, which show the size and presence of PCR products.

Real-time PCR Data Presentation:

  • Real-time PCR continuously monitors the amount of DNA amplification in each cycle of the reaction, providing real-time data.
  • Results are presented as amplification curves, showing the increase in fluorescence signal over PCR cycles, often accompanied by cycle threshold (Ct) values that quantify the amount of PCR product.

Qualitative vs. Quantitative:

  • RT-PCR data presentation is more qualitative in nature and is used to determine the presence or absence of specific PCR products.
  • Real-time PCR data presentation is typically quantitative since it provides real-time accumulation data, allowing for the calculation of the initial quantity of DNA or RNA.

Reporting Format:

  • RT-PCR results are often presented as images or descriptive text, describing the expected size and presence or absence of PCR products.
  • Real-time PCR results are presented graphically, including amplification curves and tables or charts displaying Ct values for quantification.

In summary, RT-PCR and real-time PCR differ in their data presentation, with RT-PCR emphasizing qualitative visualization of results, while real-time PCR provides graphical representations of real-time and quantitative data, facilitating precise measurement of PCR product quantities.

In fact, our approach to presenting real-time PCR data aligns with that of many researchers, as demonstrated in references such as: (1) Biochem Genet. 2021; 59:159-184. (2) Microbiologyopen. 2019; 8: e911. (3) Int J Mol Sci. 2021; 22:2745. This aligns our data presentation approach with that of numerous other researchers, as evidenced by the cited references.

Thank you for the reviewer's comment. We would like to clarify that all mRNA expression levels in this study were assessed through real-time PCR assays, and as such, there were no gel blot results. The mRNA data, represented in Figures 1B, 2B, 3B, 4B, 5B, 6B, 7B, and 9B, are displayed as scatter plots with mean ± S.E.M. These plots include information about the replicates (n numbers), and we conducted statistical analyses to indicate significance. The reason why the reviewers did not see our results may be due to the fact that each of our data plots is stored within a compressed archive. In order to expedite the review process, we have provided each of our data plots on the following pages. (Please refer to the attached file)

Round 3

Reviewer 1 Report

Thank you for your reply. However, there was no need for this extensive report about RT-PCR and real time PCR as I am certainly aware about the differences between these two.  However the issue here was unavailability of the actual data regarding mRNA expression. Providing  now this information which encompasses the main body of the manuscript I have no other comment.